# Effects of *CALR*-Mutant Type and Burden on the Phenotype of Myeloproliferative Neoplasms

**DOI:** 10.3390/diagnostics12112570

**Published:** 2022-10-23

**Authors:** Hyun-Young Kim, Yujin Han, Jun Ho Jang, Chul Won Jung, Sun-Hee Kim, Hee-Jin Kim

**Affiliations:** 1Department of Laboratory Medicine and Genetics, Samsung Medical Center, Sungkyunkwan University School of Medicine, Seoul 06351, Korea; 2Department of Laboratory Medicine, Seegene Medical Foundation, Seoul 04805, Korea; 3Division of Hematology-Oncology, Department of Medicine, Samsung Medical Center, Sungkyunkwan University School of Medicine, Seoul 06351, Korea

**Keywords:** *CALR*, type 1 mutation, type 2 mutation, mutant burden, myeloproliferative neoplasm, essential thrombocythemia, primary myelofibrosis

## Abstract

Somatic *CALR* mutations occur in approximately 70% of patients with *JAK2* V617F-negative essential thrombocythemia (ET) and primary myelofibrosis (PMF). We evaluated the effects of the *CALR* mutant type and burden on the phenotype of *CALR*-mutated myeloproliferative neoplasms (MPN). Of the 510 patients with suspected or diagnosed MPN, all 49 patients detected with *CALR* mutations were diagnosed with ET (n = 32) or PMF (n = 17). The *CALR* mutant burden was significantly higher in PMF than in ET (45% vs. 34%), and type 1-like and type 2-like mutations were detected in 49% and 51% patients, respectively. Patients with MPN and type 2-like mutation showed a significantly higher median platelet count than those with type 1-like mutation. Particularly, patients with ET and type 2-like mutation had no thrombotic events, despite higher platelet counts. The effect of *CALR* mutant burden differed depending on the mutant type. A higher mutant burden tended to be associated with a cytopenic phenotype (i.e., lower hemoglobin levels and platelet counts) in patients with the type 1-like mutation and a proliferative hematological phenotype (i.e., higher platelet and neutrophil counts) in patients with the type 2-like mutation. This study suggests that the disease phenotype of MPN may be altered through *CALR* mutant burden and mutant type.

## 1. Introduction

Calreticulin (CALR) is an endoplasmic reticulum (ER) chaperone protein involved in intracellular calcium homeostasis and the folding and quality control of newly synthesized proteins [1]. In 2013, *CALR* mutations were first identified as driver mutations in myeloproliferative neoplasm (MPN) and were found in 70–80% of patients with *JAK2* V617F-negative essential thrombocythemia (ET) and primary myelofibrosis (PMF), with characteristic clinical findings [2,3]. In ET, patients with *CALR* mutations showed male predominance, higher platelet counts, lower hemoglobin levels and leukocyte counts, and a lower risk of thrombosis than patients with *JAK2* and *MPL* mutations [4]. In patients with PMF, *CALR* mutations were associated with a younger age and higher platelet counts and were less likely to be related to anemia and leukocytosis [5]. These patients also showed favorable survival compared to patients with *JAK2*-mutated and triple-negative PMF [5].

*CALR* mutations typically result in a +1-bp frameshift through insertions and/or deletions in exon 9, resulting in a mutant CALR with a novel C-terminus comprising varying amounts of negatively charged amino acids of wild-type CALR [2,3]. A 52-bp deletion (type 1; c.1099_1150del [p.Leu367Thrfs*46]) and a 5-bp insertion (type 2; c.1154_1155insTTGTC [p.Lys385Asnfs*47]) are the most common mutations, accounting for 85% of *CALR* mutations. Type 1 mutation eliminates almost all of the negatively charged amino acids, whereas type 2 mutation retains approximately half of the negatively charged amino acids [2]. Other *CALR* mutations can also be classified into type 1-like and type 2-like mutations according to the extent of residual negatively charged amino acids [6,7]. 

To date, several studies have reported different phenotypic and prognostic effects, depending on the *CALR* mutant type (type 1/type 1-like vs. type 2/type 2-like) [6,7,8,9]. However, the phenotypic characteristics of each *CALR* mutant type differ slightly between studies [6,7]. Additionally, there are limited studies on the effect of the *CALR* mutant burden on the disease phenotype. In this study, we evaluated the differences in clinical and hematological phenotypes according to the *CALR* mutant types and the effect of *CALR* mutant burden on the phenotype in *CALR*-mutated MPN. 

## 2. Materials and Methods

### 2.1. Study Patients

We retrospectively reviewed the results of a *CALR* mutation analysis conducted in patients with suspected or diagnosed MPN from July 2015 to May 2020, and patients with *CALR* mutations were included in the study. All patients were diagnosed or revised according to the 2016 World Health Organization (WHO) classification of myeloid neoplasms [10], and the patients’ clinical and laboratory information including diagnosis, sex, age, complete blood cell counts, lactate dehydrogenase levels, splenomegaly, history of thrombosis, and survival were obtained from electronic medical records. This study was conducted in accordance with the Declaration of Helsinki. The Institutional Review Board (IRB) of the Samsung Medical Center (SMC), Seoul, Korea, approved this study (SMC IRB No. 2021-07-174) and waived the need for informed consent for retrospective data collection and review. 

### 2.2. CALR Mutation Analysis

Genomic DNA was isolated from peripheral blood or bone marrow aspirates using a Wizard Genomic DNA Purification Kit (Promega, Madison, WI, USA) according to the manufacturer’s instructions. *CALR* mutation analysis was performed sequentially via fragment analysis and direct sequencing. PCR was performed with primers that amplify exon 9 of *CALR* (forward primer 5′-CTGGTCCTGGTCCTGATGTC-3′ and reverse primer 5′-CGAACCAGCCTGGAAAAA-3′) using a Thermal Cycler 9700 (Applied Biosystems, Foster City, CA, USA). Fragment analysis was performed by marking the forward primer with a fluorescent dye (5′-FAM). The PCR product size was measured with ABI Prism 3130xl Genetic Analyzer (Applied Biosystems) using GeneMapper Software 4.0 (Applied Biosystems). Direct sequencing was performed using an ABI Prism 3130xl Genetic Analyzer with a BigDye Terminator Cycle Sequencing Ready Reaction Kit (Applied Biosystems). The *CALR* mutant burden (%) was calculated as the area of mutant allele/[area of mutant allele + area of wild-type allele] × 100. *CALR* mutations were classified into type 1-like or type 2-like mutation groups on the basis of the predicted effect on the C-terminus of CALR, as previously reported [6]. 

### 2.3. Statistical Analysis

Categorical variables were compared using the chi-square or Fisher’s exact test, and numerical variables were compared using the Mann–Whitney test or two-sample *t*-test, as appropriate. Spearman correlation analysis was performed to evaluate the relationship between *CALR* mutant burden and other laboratory results. *p*-value < 0.05 was considered statistically significant. Statistical analyses were performed using the IBM SPSS version 27 (IBM Corp., Armonk, NY, USA).

## 3. Results

### 3.1. Patient Characteristics

Among the 510 patients screened, 49 (9.6%) had *CALR* mutations, and their characteristics are summarized in Table 1. The median age of the patients was 56 years, and 19 patients (39%) were male. All patients with *CALR* mutations were diagnosed either as ET (n = 32, 65%) or PMF (n = 17, 35%), and the median *CALR* mutant burden was significantly higher in patients with PMF than in those with ET (45% vs. 34%, *p* = 0.001). In addition to patients with *CALR* mutation, 150 (29.4%) had a *JAK2* V617F mutation, and none had both mutations.

### 3.2. CALR Mutant Types and Their Effects on the Phenotype

The identified *CALR* mutations are summarized in Table 2. Type 1 and type 2 *CALR* mutations were detected in 20 (41%) and 22 (45%) patients, respectively, and three and two unique *CALR* mutations were classified as type 1-like and type 2-like mutations, respectively. Two patients had 16-bp and 46-bp deletion mutations, identified only by fragment analysis because of low *CALR* mutant burden, and were excluded from further analysis for *CALR* mutant type.

Altogether, 23 and 24 patients with type 1-like (including type 1) and type 2-like (including type 2) *CALR* mutations were identified, respectively, and their clinical and laboratory characteristics were compared (Table 3). Although there were no significant differences in leukocyte count, hemoglobin level, and *CALR* mutant burden, the median platelet count was significantly higher in patients who had MPN with type 2-like *CALR* mutation than in those with type 1-like *CALR* mutation (892 × 10^9^/L vs. 491 × 10^9^/L, *p* < 0.001); similarly, ET seemed to be more frequent than PMF (79% vs. 21%, *p* = 0.051) in those with type 2-like *CALR* mutation. In contrast, the median lactate dehydrogenase level (580 IU/L vs. 413 IU/L, *p* = 0.020) and frequency of splenomegaly (44% vs. 17%, *p* = 0.045) were higher in patients with type 1-like *CALR* mutations than in those with type 2-like *CALR* mutations. 

### 3.3. Effects of CALR Mutant Type on Phenotype in ET and PMF

We further compared the clinical and laboratory characteristics of the patients with ET and PMF according to the *CALR* mutant type (Table 4). In patients with ET, the median leukocyte count (6.66 × 10^9^/L vs. 8.15 × 10^9^/L, *p* = 0.074) and hemoglobin level (12.2 g/dL vs. 13.8 g/dL, *p* = 0.059) were lower in patients with type 2-like *CALR* mutation than in those with type 1-like *CALR* mutation, whereas the median platelet count (851 × 10^9^/L vs. 705 × 10^9^/L, *p* = 0.092) was higher in patients with type 2-like *CALR* mutations. In particular, 19 patients with ET and type 2-like *CALR* mutation did not show any thrombotic events (*p* = 0.049). In patients with PMF, the median hemoglobin level (13.5 g/dL vs. 8.9 g/dL, *p* = 0.027) and platelet count (1513 × 10^9^/L vs. 210 × 10^9^/L, *p* = 0.003) were higher in patients with type 2-like *CALR* mutation than in those with type 1-like *CALR* mutation. In particular, the platelet count in patients with PMF and type 2-like *CALR* mutation was higher than that in patients with ET and the same *CALR* mutant type (*p* = 0.043). However, these findings may have limited implications because of the small number of events as well as the small number of patients with ET and PMF belonging to each *CALR* mutant type, especially those with PMF and type 2-like mutation. There was no difference in *CALR* mutant burden according to the mutant type in patients with ET (*p* = 0.292) and PMF (*p* = 0.910).

### 3.4. Correlation between CALR Mutant Burden and Hematological Phenotype in Type 1-like and Type 2-like CALR Mutations

In all patients with *CALR* mutations, as the mutant burden increased, leukocyte count increased and hemoglobin level decreased (Figure 1). We further performed a correlation analysis between the *CALR* mutant burden and laboratory variables according to the *CALR* mutant type (Figure 2 and Appendix A). In patients with type 1-like mutation, the *CALR* mutant burden was negatively correlated with hemoglobin level (*r* = −0.705, *p* < 0.001) and platelet count (*r* = −0.570, *p* = 0.004). In contrast, the *CALR* mutant burden was positively correlated with absolute neutrophil count (*r* = 0.409, *p* = 0.047) and platelet count (*r* = 0.457, *p* = 0.025) in patients with type 2-like *CALR* mutation.

### 3.5. Correlation of CALR Mutant Burden and Hematological Phenotype in ET and PMF

In patients with ET, hemoglobin levels decreased with increasing *CALR* mutant burden (*r* = −0.506, *p* = 0.003) (Figure 3). There was no significant correlation between *CALR* mutant burden and leukocyte or platelet counts in these patients (Appendix A). Conversely, in patients with PMF, leukocyte count was positively correlated with *CALR* mutant burden (*r* = 0.489, *p* = 0.046), whereas hemoglobin level and platelet count were not significantly correlated (Appendix A).

## 4. Discussion

The impact of *CALR* mutant type on clinical and laboratory phenotypes has been demonstrated in several studies; however, there were some inconsistent findings [6,7,8,9,11]. In our study, type 1-like *CALR* mutation was predominant in PMF compared with ET, and patients with MPN and type 2-like mutation were characterized by profound thrombocytosis than those with type 1-like mutation. These findings are consistent with those of previous studies showing that patients who have MPN with type 1-like *CALR* mutation are associated with a myelofibrosis phenotype and those with type 2-like mutation are mainly associated with an ET phenotype [6,9]. 

Depending on the MPN subtype, patients with ET and type 1-like *CALR* mutation have a higher risk of myelofibrosis transformation than those with type 2-like mutation. Patients with ET and type 2-like mutation are characterized by a younger age and higher platelet count than those with type 1-like mutation [6,9,11]. In particular, patients with ET and *CALR* mutation are known to have a lower risk of thrombosis compared to those with JAK2 V617F mutation [4,12]; however, this seems to be different depending on the *CALR* mutant type. In our study, thrombotic events were observed in 9% and 18% of patients with ET and PMF carrying *CALR* mutation, respectively, which were not significantly different from those carrying *JAK2* V617F mutation (13% [7/55], *p* = 0.636; 17% [5/29], *p* = 0.972, respectively) (data not submitted). However, although study patients were small for a complete analysis, there were no thrombotic events in 19 patients with ET and type 2-like *CALR* mutation, suggesting a low risk of thrombosis in these patients. Similarly, Pietra et al. showed that patients with ET carrying type 2-like *CALR* mutation had a lower risk of thrombosis compared with those carrying *JAK2* V617F mutation, but not those carrying type 1-like *CALR* mutation [6]. They reported that none of the 84 patients with ET and type 2-like mutation had thrombosis.

In patients with PMF, Tefferi et al. showed that type 2-like *CALR* mutation is associated with a higher dynamic international prognostic scoring system plus score, *EZH2* mutations, leukocytosis, higher peripheral blast percentage, palpable spleen size, and significantly shorter survival [7,8]. Cabagnols et al. also showed that patients with PMF and type 2 *CALR* mutations were characterized by a higher median platelet count, highlighting that no patients had a platelet count below 100 × 10^9^/L [9]. In contrast, Pietra et al. demonstrated no significant differences in hematological phenotypes according to the *CALR* mutant type in patients with PMF [6]. In our study, although statistical power was limited due to the small number of patients, patients with PMF and type 2-like *CALR* mutation seemed to have a proliferative hematological phenotype of higher hemoglobin levels and platelet counts, helping to distinguish them from those with type 1-like mutation. In particular, while patients with PMF and type 2-like mutation mimicked the ET phenotype in terms of thrombocytosis, they tended to exhibit a higher leukocyte count and *CALR* mutant burden than patients with ET and the same *CALR* mutant type.

Mutant *CALR* has been suggested to induce MPN via interactions with the thrombopoietin receptor (MPL) [13,14]. Normally, the last domain of CALR consists of a string of negatively charged amino acid residues responsible for Ca^2+^ buffering and C-terminus ER retention signal (KDEL) [14]. In contrast, mutant CALR lacks the KDEL and is rich in positively charged amino acids [2], and the oncogenicity of mutant CALR is thought to depend on the positive charge of the C-terminus, which is necessary for the physical interaction between mutant CALR and MPL [14]. Particularly, a previous study demonstrated a larger Ca^2+^ release from the ER in cultured megakaryocytes of patients with MPN and type 1 *CALR* mutations than in patients carrying either the *JAK2* mutation or type 2 *CALR* mutation, suggesting that phenotypic differences between type 1-like and type 2-like *CALR* mutations may be related to the greater loss of calcium-binding sites in patients with type 1-like mutation [6].

The *CALR* mutant burden is also considered to influence the disease phenotype of MPN, similar to the phenotypic impact of the *JAK2* mutant burden [15,16,17,18,19]. Specifically, a higher *CALR* mutant burden has been observed in patients with PMF than in those with ET [7,9,20,21]. Li et al. reported a median *CALR* mutant burden of 38% in 28 patients with PMF compared with 29% in 124 patients with ET [20]. Gángó et al. reported a median *CALR* mutant burden of 43% and 50% in patients with ET and PMF, respectively [21]. In our study, the median *CALR* mutant burden was 34% and 45% in ET and PMF, respectively, and a high mutant burden of >45% was more frequent in PMF than in ET (59% vs. 9%, *p* = 0.006). 

Additionally, our study suggested that a higher *CALR* mutant burden is associated with advanced disease phenotypes, such as leukocytosis and anemia in all patients with MPN. In particular, we noted the different phenotypic effects of the *CALR* mutant burden depending on the *CALR* mutant type. A higher mutant burden tended to be associated with a cytopenic phenotype (i.e., lower hemoglobin levels and platelet counts) in patients with type 1-like *CALR* mutation, whereas it seemed to be associated with a proliferative hematological phenotype (i.e., higher platelet and neutrophil counts) in patients with type 2-like *CALR* mutation. To our knowledge, this is the first study to suggest that the phenotypic effect of *CALR* mutant burden may differs in each *CALR* mutant type. Moreover, we showed the different phenotypic effects of *CALR* mutant burden in ET and PMF. In ET, hemoglobin levels decreased as the *CALR* mutant burden increased, which was consistent with previous studies showing lower hemoglobin levels in patients with ET with a higher *CALR* mutant burden [11,21]. However, Bertozzi et al. showed no correlation between *CALR* mutant burden quartile and laboratory parameters [22]. Instead, they showed that the highest mutant burden quartile was associated with poorer survival and more frequent evolution into myelofibrosis and leukemia. In PMF, although the number of patients and range of mutant burden were limited, leukocyte counts appeared to increase with the increasing *CALR* mutant burden.

Our study had some limitations. First, a small number of patients were included in the study. Second, the effects of other concomitant somatic mutations were not considered. Finally, a comparison with *JAK2*-mutated MPN was not performed. 

In conclusion, this study suggests that the disease phenotype may be altered through *CALR* mutant burden as well as mutant type in patients with *CALR*-mutated MPN, and in particular, the effect of *CALR* mutant burden may differ depending on the mutation type. Further studies with a larger number of patients are needed to validate our results.

## Figures and Tables

**Figure 1 diagnostics-12-02570-f001:**
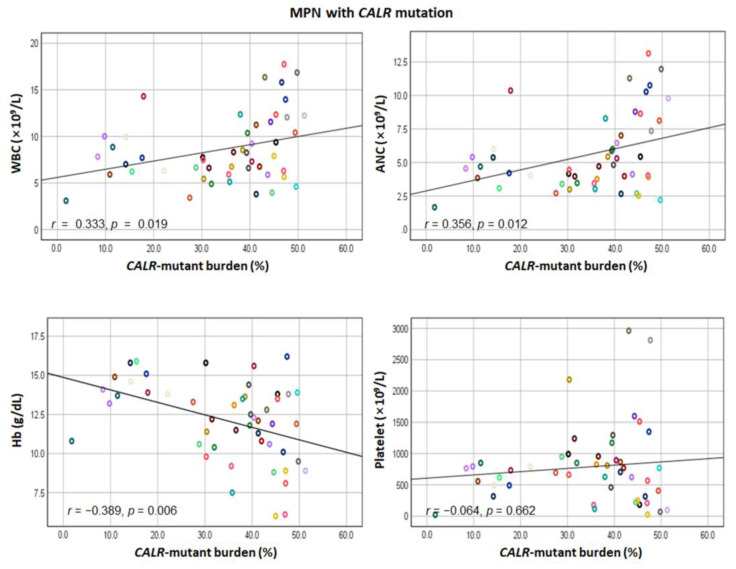
Scatter plots between *CALR*-mutant burden and white blood cell (WBC) count, absolute neutrophil count (ANC), hemoglobin (Hb) level, and platelet count in 49 MPN patients with *CALR* mutation. Circles of the same color indicate the same patient.

**Figure 2 diagnostics-12-02570-f002:**
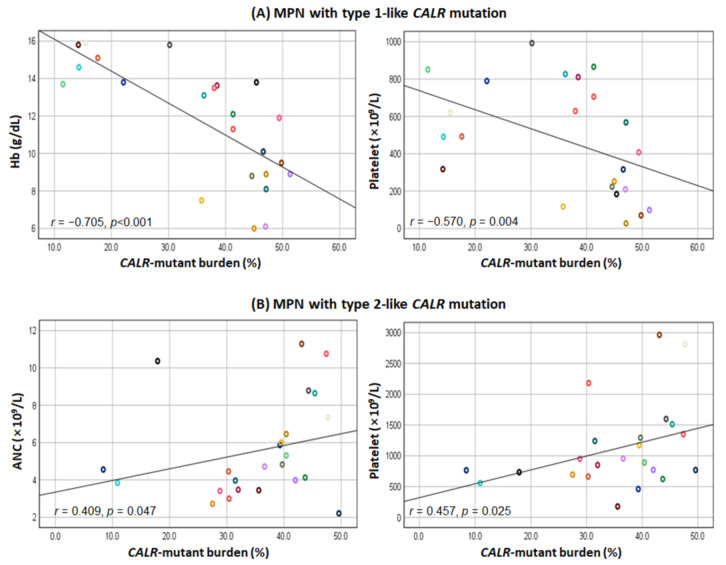
(**A**) Scatter plots between *CALR*-mutant burden and hemoglobin (Hb) level and platelet count in MPN patients with type 1-like *CALR* mutation; (**B**) scatter plots between *CALR*-mutant burden and absolute neutrophil count (ANC) and platelet count in MPN patients with type 2-like *CALR* mutation. Circles of the same color in each group indicate the same patient.

**Figure 3 diagnostics-12-02570-f003:**
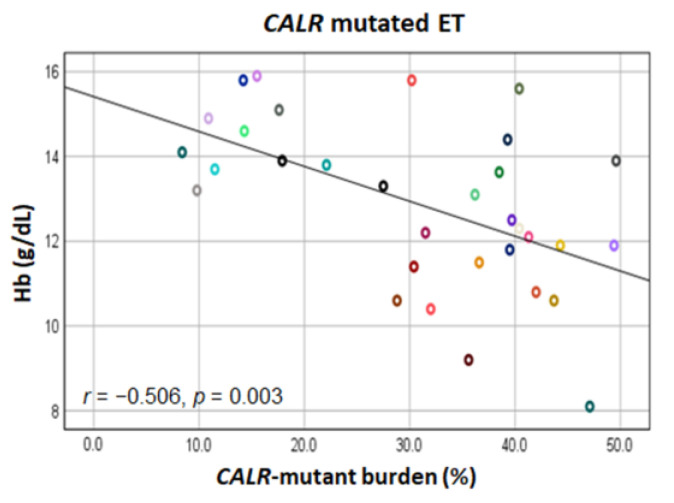
Scatter plots between *CALR*-mutant burden and hemoglobin (Hb) level in patients with *CALR*-mutated ET.

**Table 1 diagnostics-12-02570-t001:** Characteristics of patients with *CALR* mutation.

	Total (n = 49)	ET (n = 32)	PMF (n = 17)	*p*-Value
Age, years (IQR)	56 (47–64)	54 (47–63)	61 (46–67)	0.366
Male, n (%)	19 (39%)	11 (34%)	8 (47%)	0.386
WBC, ×10^9^/L (IQR)	7.76 (6.08–10.84)	7.51 (6.25–9.78)	9.38 (5.40–13.17)	0.324
ANC, ×10^9^/L (IQR)	4.70 (3.63–7.19)	4.63 (3.79–5.98)	5.44 (2.87–10.04)	0.515
AMC, ×10^9^/L (IQR)	0.56 (0.40–0.73)	0.50 (0.35–0.66)	0.67 (0.40–0.98)	0.085
Hb, g/dL (IQR)	12.3 (10.5–13.9)	13.2 (11.6–14.3)	10.1 (8.9–13.5)	0.007
Platelets, ×10^9^/L (IQR)	734 (363–923)	793 (581–937)	252 (108–1028)	0.019
LDH, IU/L (IQR)	434 (370–607)	399 (301–462)	581 (434–1014)	0.011
Splenomegaly, n (%)	15 (31%)	5 (16%)	10 (59%)	0.002
CALR mutant burden, %	35 (28–45)	34 (18–40)	45 (40–47)	0.001
Follow-up duration, months	47 (27–57)	41 (26–55)	49 (27–69)	0.535
Thrombotic events, n (%)	6 (12%)	3 (9%)	3 (18%)	0.405
Leukemic transformation, n (%)	1 (2%)	0 (0%)	1 (6%)	0.347
Deceased, n (%)	4 (8%)	1 (3%)	3 (18%)	0.114

Abbreviations: ET, essential thrombocythemia; PMF, primary myelofibrosis; IQR, interquartile range; WBC, white blood cells; ANC, absolute neutrophil counts; AMC, absolute monocyte counts; Hb, hemoglobin; LDH, lactate dehydrogenase.

**Table 2 diagnostics-12-02570-t002:** Identified *CALR* mutations and their mutant types.

*CALR* Mutant Type ^a^ (COSMIC ID)	Nucleotide Change	Amino Acid Change	No. of Patients (%)		
			ET	PMF	Total
Type 1-like mutation					
Type 1 (COSM1738055)	c.1099_1150del	p.Leu367Thrfs*46	11 (55%)	9 (45%)	20 (41%)
Type 3 (COSM1738150)	c.1100_1145del	p.Leu367Glnfs*48	1 (100%)	-	1 (2%)
Type 7 (COSM1738343)	c.1103_1154del	p.Lys368Argfs*45	-	1 (100%)	1 (2%)
n.d. (COSM3355766)	c.1103_1148del	p.Lys368Argfs*47	-	1 (100%)	1 (2%)
Type 2-like mutation					
Type 2 (COSM1738056)	c.1154_1155insTTGTC	p.Lys385Asnfs*47	18 (82%)	4 (18%)	22 (45%)
Type 33 (COSM1738355)	c.1154_1155insATGTC	p.Glu386Cysfs*46	-	1 (100%)	1 (2%)
Type 35 (COSM1738356)	c.1154delinsTTTGTC	p.Lys385Ilefs*47	1 (100%)	-	1 (2%)

^a^ Specific *CALR* mutant type was designated based on a report by Klampfl et al. [2]. Abbreviations: COSMIC, Catalogue Of Somatic Mutations In Cancer; ET, essential thrombocythemia; PMF, primary myelofibrosis; n.d., not designated.

**Table 3 diagnostics-12-02570-t003:** Comparison of patients' clinical and laboratory characteristics according to the *CALR* mutant types.

	*CALR*-Mutated MPN		
	Type 1-like Mutation (n = 23)	Type 2-like Mutation (n = 24)	*p*-Value
Age, years (IQR)	61 (47–65)	51 (47–61)	0.250
Male, n (%)	11 (48%)	6 (25%)	0.104
WBC, ×10^9^/L (IQR)	7.90 (6.31–11.25)	7.40 (5.92–11.3)	0.537
ANC, ×10^9^/L (IQR)	4.70 (3.77–8.13)	4.64 (3.57–7.14)	0.941
AMC, ×10^9^/L (IQR)	0.59 (0.40–0.79)	0.53 (0.33–0.68)	0.317
Hb, g/dL (IQR)	12.1 (8.9–13.8)	12.4 (11.0–13.9)	0.383
Platelets, ×10^9^/L (IQR)	491 (210–790)	892 (705–1336)	<0.001
LDH, IU/L (IQR)	580 (426–1112)	413 (268–483)	0.020
Subtypes of MPN			
ET, n (%)	12 (52%)	19 (79%)	0.051
PMF, n (%)	11 (48%)	5 (21%)	
Splenomegaly, n (%)	10 (44%)	4 (17%)	0.045
Thrombotic events, n (%)	5 (22%)	1 (4%)	0.097
*CALR*-mutant burden, %	41 (22–47)	39 (30–44)	0.503

Abbreviations: MPN, myeloproliferative neoplasm; ET, essential thrombocythemia; PMF, primary myelofibrosis; IQR, interquartile range; WBC, white blood cells; ANC, absolute neutrophil counts; AMC, absolute monocyte counts; Hb, hemoglobin; LDH, lactate dehydrogenase.

**Table 4 diagnostics-12-02570-t004:** Comparison of patients' clinical and laboratory characteristics according to the *CALR* mutant types in ET and PMF.

	ET			PMF			Type 1-like Mutation: ET vs. PMF	Type 2-like Mutation: ET vs. PMF
	Type 1-like Mutation (n = 12)	Type 2-like Mutation (n = 19)	*p*-Value	Type 1-like Mutation (n = 11)	Type 2-like Mutation (n = 5)	*p*-Value ^a^	*p*-Value	*p*-Value ^a^
Age, years (IQR)	58 (46–63)	51 (47–61)	0.715	63 (48–67)	51 (42–72)	0.395	0.323	0.859
Male, n (%)	6 (50%)	4 (21%)	0.127	5 (45%)	2 (40%)	1.000	0.827	0.568
WBC, ×10^9^/L (IQR)	8.15 (6.83–10.31)	6.66 (5.89–8.32)	0.074	7.90 (5.13–12.4)	12.35 (9.78–15.17)	0.193	0.667	0.008
ANC, ×10^9^/L (IQR)	5.04 (4.07–6.77)	4.13 (3.45–5.87)	0.201	4.04 (2.70–9.79)	8.65 (5.91–11.03)	0.126	0.538	0.014
AMC, ×10^9^/L (IQR)	0.56 (0.44–0.71)	0.45 (0.32–0.60)	0.105	0.67 (0.40–0.98)	0.68 (0.61–1.14)	0.467	0.854	0.009
Hb, g/dL (IQR)	13.8 (12.4–15.6)	12.2 (10.8–13.9)	0.059	8.9 (7.50–11.30)	13.5 (11.3–15.0)	0.027	0.002	0.455
Platelets, ×10^9^/L (IQR)	705 (492–845)	851 (695–1173)	0.092	210 (99–316)	1513 (1007–2888)	0.003	0.001	0.043
LDH, IU/L (IQR)	402 (382–422)	399 (279–477)	1.000	775 (542–1167)	428 (245–500)	0.039	0.046	0.800
Splenomegaly, n (%)	3 (25%)	2 (11%)	0.350	7 (64%)	2 (40%)	0.596	0.100	0.179
Thrombotic events, n (%)	3 (25%)	0 (0%)	0.049	2 (18%)	1 (20%)	1.000	1.000	0.208
*CALR*-mutant burden, %	26 (15–41)	37 (29–40)	0.292	45 (41–47)	45 (37–48)	0.910	0.008	0.051

^a^ Due to the small number of patients with PMF and type 2-like mutation (n = 5), the *p*-values comparing these patient group to other patient groups may have very limited statistical power. Abbreviations: ET, essential thrombocythemia; PMF, primary myelofibrosis; IQR, interquartile range; WBC, white blood cells; ANC, absolute neutrophil counts; AMC, absolute monocyte counts; Hb, hemoglobin; LDH, lactate dehydrogenase.

## Data Availability

The data that support the findings of this study are available from the corresponding author upon reasonable request.

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
