# Peer review of "Effects of CALR-Mutant Type and Burden on the Phenotype of Myeloproliferative Neoplasms"

_diagnostics, 2022, doi:10.3390/diagnostics12112570_

Round 1

Reviewer 1 Report

The paper by Kim et al profiles the CARL mutations harbored by 510 patients with myeloproliferative neoplasms. The paper describes that 49 patients, out of the 510 patients included in the study, present mutation in the CARL gene. The CARL mutations were both of Type I and Type II and, most interesting, one of the Type 1-like mutations appears new. The authors may want to clarify this impression. The paper presents several correlations between type of CARL mutation, allele burden and disease phenotype only some of which may have the power to be meaningful. The authors should be very carefully not to overstate conclusions which are based on  low number of patients. In conclusion, in general, this paper is robust and contributes to assess the frequency of CARL mutations in MPN patients worldwide.

Major Comments

- In general, the power of the analyses (total number of patients included in the groups) is robust. From a power point of view, It is also good that the correlations described in Table 3 are done on the MPN population as a hole (49 patients analyzed).  

- The possible identification of a new Type 1 mutation is a strength. The authors may want to clarify this point.

-   Page 4, Lines 124-125. With a total of 49 CARLpos patients, it is unclear whether the study has the power to assess differences in frequency of type 1 and type 2 mutations in ET vs PMF. With 5 thrombotic events total, it is also almost impossible to make any conclusion on associations between type of CARL mutation and frequency of thrombosis in ET and PMF. These statements, and similar statements with low power, should be tuned down.  

-         -  Table 4. The number of patients included is OK for the ET Type 1, ET Type 2 and PMF type 2 groups. It is too low to provide power in the PMF type 2 group (5 patients).  The p values for the PMF groups and for the ET type 2 vs PMF type 2, although below p.05, may not have significant power. The Table and the text should be revised to clearly indicate this limitation.

-          - I may have missed it, but how is the frequency of the thrombotic events in the CARL pos vs the JAK2 pos patients? The authors should have all the data to assess this point.

-          - Figure 1. This Figure appears foggy. Its quality needs improvement. As presently designed, the positive correlations are dispersed among the negative ones. It is suggested to move the negative correlations in the Supplementary. It would be interested to symbol code the values from the same patients in the various panels. This would allow to identify if the correlations are conserved on an individual patient basis.

-          - Figure 2. Same comments than for Figure 1. Consider symbol or color coding the individual patients and moving correlations which are not significant in the Supplementary.

Minor comments

- Abstract: Please clarify here the mutations harbored by the 461 patients who did how had CARL mutations.

Reviewer 2 Report

This study analyzes the characteristics of the MPNs depending on the type of mutation in CALR. This has already been done with larger series of patients, so in that sense it is not original. However, it is true that the previous findings are controversial and that they provide different information. One of the most interesting findings already described in previous studies has been finding a low incidence of thrombosis when the mutation is in CALR type 2. This finding could have repercussions in clinical practice. In that sense, I think it should have greater importance in the discussion

Round 2

Reviewer 1 Report

All my comments were appropriately addressed. I am surprised that the number of double negative patients is so high (as many as 150). This would indicate a frequency of Mpl mutations plus triple negative greater than that usually reported. The authors may want to add a comment in this regard in the galleys. But all good from my part.